# Multi-Level Thresholding Based on Composite Local Contour Shannon Entropy Under Multiscale Multiplication Transform

**DOI:** 10.3390/e27050544

**Published:** 2025-05-21

**Authors:** Xianzhao Li, Yaobin Zou

**Affiliations:** 1Hubei Key Laboratory of Intelligent Vision Based Monitoring for Hydroelectric Engineering, China Three Gorges University, Yichang 443002, China; 202208540021013@ctgu.edu.cn; 2College of Computer and Information Technology, China Three Gorges University, Yichang 443002, China

**Keywords:** Shannon entropy, multi-level thresholding, multiscale multiplication transform, composite contour

## Abstract

Image segmentation is a crucial step in image processing and analysis, with multi-level thresholding being one of the important techniques for image segmentation. Existing approaches predominantly rely on metaheuristic optimization algorithms, which frequently encounter local optima stagnation and require extensive parameter tuning, thereby degrading segmentation accuracy and computational efficiency. This paper proposes a Shannon entropy-based multi-level thresholding method that utilizes composite contours. The method selects appropriate multiscale multiplication images by maximizing the Shannon entropy difference and constructs a new Shannon entropy objective function by dynamically combining contour images. Ultimately, it automatically determines multiple thresholds by integrating local contour Shannon entropy. Experimental results on synthetic images and real-world images with complex backgrounds, low contrast, blurred boundaries, and unbalanced sizes demonstrate that the proposed method outperforms six recently proposed multi-level thresholding methods based on the Matthew’s correlation coefficient, indicating stronger adaptability and robustness for segmentation without requiring complex parameter tuning.

## 1. Introduction

Image segmentation is a critical step in image processing and analysis, as it effectively simplifies the representation of images and facilitates subsequent quantitative analysis. Currently, representative segmentation techniques include image thresholding [1], active contour models [2], clustering methods [3], region growing [4], and deep learning approaches [5]. Image thresholding methods, due to their simplicity, efficiency, and ease of implementation, have gained extensive application across various fields, such as industrial non-destructive testing [6], ship target monitoring [7], infrared-based power equipment fault diagnosis [8], agricultural disease monitoring [9], and brain tumor identification [10].

Image thresholding methods are primarily categorized into two types: global and local methods. Global thresholding methods select thresholds based on the statistical properties of the entire image. These methods are further subdivided into bi-level and multi-level thresholding approaches, depending on the number of thresholds used. Bi-level thresholding methods classify image pixels into target and background by setting a single global threshold. This approach is suitable for images with a clear contrast between the target and background, consisting of only two classes. In contrast, multi-level thresholding methods employ multiple thresholds to segment the image into several regions, making them more suitable for scenarios where the grayscale distribution of the target and background is more complex.

The most straightforward method for finding multiple thresholds is an exhaustive search. However, this approach presents significant challenges in determining the optimal thresholds, as increasing the number of thresholds leads to exponential growth in computational complexity. In the early stages, Chen et al. addressed this issue by reducing the search space for optimal thresholds through grayscale quantization and introduced a fast algorithm based on two-dimensional entropy [11]. This approach reduced the computational complexity from $O(L^{4})$ to $O(L^{8/3})$, where L represents the maximum grayscale. Wu et al. later introduced a fast recursive algorithm [12] that employs mathematical recursion to generate intermediate data for two-dimensional entropy, thereby further reducing the complexity to $O(L^{2})$. Inspired by Wu et al.’s work [12], many studies have applied the algorithmic design of mathematical recursion to other objective functions for multi-level thresholding [13]. Liao et al. proposed a modified between-class variance method [14] that enables the easy computation of intermediate data through mathematical recursion, which can then be stored in a lookup table for rapid retrieval. Although Liao et al.’s method reduces redundant computations, it does not completely overcome the challenge of exponential computational complexity. Yin and Chen proposed a fast iterative algorithm that reduces complexity to $O(kN^{2}L)$, where k is the number of iterations and N is the maximum number of thresholds, by iteratively searching for the optimal multiple thresholds within the subranges of thresholds [15]. Shang et al. proposed a gradient-based multi-level thresholding method that provides significant efficiency advantages. However, this method’s applicability is restricted to specific objective functions, such as Kapur’s entropy and Otsu’s inter-class variance, and it is not currently generalizable to other objective functions [16].

Due to their efficiency, flexibility, and global search capabilities, metaheuristic algorithms have been integrated into multi-level thresholding methods by numerous studies in recent years. Xing and He introduced a multi-objective emperor penguin optimizer based on clonal selection, applying this method to the multi-level thresholding of infrared power transformer images [8]. This method demonstrates high segmentation accuracy and robustness, especially in solving the fault diagnosis of power transformer segmentation. However, its main limitation is higher time complexity. The fitness function design and the optimization ability are the main factors affecting CPU runtime. Song et al. proposed a modified snake optimizer to optimize multi-level thresholding for agricultural disease images [9]. The algorithm demonstrates efficiency and robustness in multi-level thresholding by dynamically adjusting parameters and balancing global and local search capabilities, showing promising potential for agricultural disease image segmentation. However, it has some limitations, including high computational complexity, challenging parameter tuning, and unverified generalization ability. Wang et al. proposed an improved whale optimization algorithm based on Latin hypercube sampling initialization and applied it to the multi-level thresholding for COVID-19 X-ray images [17]. This method significantly enhances the robustness and accuracy of segmentation for COVID-19 X-ray images by combining multiple optimization strategies. However, it has limitations, including high computational complexity and challenges in parameter tuning. Wang et al. developed a whale optimization algorithm that combines mutation and similarity removal to address grayscale image segmentation tasks with the number of thresholds ranging from two to eight [18]. Nie et al. [19] proposed a multi-level thresholding method for crack image segmentation based on the minimum arithmetic–geometric divergence and an improved particle swarm optimization algorithm. This method enhances algorithm diversity through local stochastic perturbation, and experimental results demonstrate its superiority over several other advanced multi-level thresholding methods across various evaluation metrics [19]. Rodriguez-Esparza et al. proposed an efficient methodology for multi-level thresholding using the Harris hawks optimization (HHO) algorithm and the minimum cross-entropy as a fitness function [20]. This method has been tested on a benchmark set of reference images, including the Berkeley segmentation database and medical images from digital mammography. The experimental results, validated through statistical analysis, demonstrate that this method yields efficient and reliable results in terms of quality, consistency, and accuracy compared to other methods. Wang et al. introduced a reinforcement learning-based golden jackal optimization algorithm, named QLGJO, to segment CT images for the diagnosis of COVID-19 [21]. This approach represents the first instance of combining reinforcement learning with meta-heuristics in a segmentation problem. The strategy effectively addresses the limitation of the original algorithm, which tends to fall into local optima. Fu et al. proposed a multi-level thresholding method based on an improved northern goshawk optimization (INGO) algorithm [22]. By integrating cubic chaotic optimization and lens imaging-based reverse learning strategies, this approach enhances population diversity, expands the search space, and optimizes initial solutions, enabling the INGO algorithm to explore potential optimal solutions more effectively. Jena et al. developed an enhanced barnacle mating optimization (EBMO) algorithm for multi-level thresholding, improving the original method by incorporating an additional Gaussian mutation strategy and a random flow towards the best solution steps [23]. Mittal and Saraswat proposed a new non-local means 2D histogram and introduced a novel variant of the gravitational search algorithm, known as the exponential Kbest gravitational search algorithm, to identify optimal thresholds [24]. Additionally, a 2D Renyi entropy has been redefined for multi-level thresholding optimization. This method was tested on the Berkeley segmentation dataset and benchmarked using both subjective and objective assessments. The experimental results confirm that this method outperforms other 2D histogram-based image thresholding methods across the majority of performance parameters. Wang et al. proposed a novel tuna swarm optimization algorithm enhanced with a Sigmoid nonlinear weights strategy, quadratic interpolation, and elite swarm genetic operators [25]. This method outperforms traditional techniques such as Otsu and MCET in terms of convergence and global optimization capabilities for rice plant image segmentation, demonstrating its significant potential in improving agricultural image analysis and yield prediction. While metaheuristic algorithms can address the issue of high computational cost in multi-level thresholding, most of these algorithms are prone to becoming trapped in local optima and have a large number of parameters, which can still affect the thresholding results and efficiency [26,27,28].

Unlike existing research that introduces metaheuristic algorithms to compute multiple thresholds, this paper proposes a new multi-level thresholding based on composite local contour Shannon entropy under multiscale multiplication transform (CLCSE). Based on a novel theoretical analysis about Shannon entropy difference, the proposed method first selects an appropriate multiscale multiplication image by maximizing the Shannon entropy difference. Subsequently, a new Shannon entropy objective function is constructed using the multiscale multiplication image as a guide, along with dynamically changing contour images. Finally, the automatic selection of multiple thresholds is achieved by combining the local contour Shannon entropy.

The rest of the paper is organized as follows: Section 2 proposes and demonstrates the properties of the Shannon entropy difference. Section 3 introduces and analyzes the computation of multiscale multiplication images, which is guided by maximizing the difference in Shannon entropy. Section 4 constructs a new Shannon entropy objective function. Section 5 introduces the selection method of multiple thresholds based on the computation of composite local contour Shannon entropy. Section 6 outlines the algorithmic steps of the proposed CLCSE method. Section 7 compares and analyzes the experimental results of seven multi-level thresholding methods. Finally, Section 8 draws some conclusions and suggests directions for future research.

## 2. The Definition of Shannon Entropy Difference

For any given grayscale histogram, a grayscale l can be used to divide the histogram into two parts: one part to the left and one part to the right (see Figure 1). Assuming the left histogram contains a total of $\psi_{-}$ grayscale levels with a total frequency represented by $\mu_{-}$, and there is a discrete probability distribution for the grayscale levels $q_{i}(1\leq i\leq \psi_{-})$ such that $\sum_{i=1}^{\psi_{-}}q_{i}=1$, the right histogram has $\psi_{+}$ grayscale levels with a total frequency of $\mu_{+}$, and a discrete probability distribution of the grayscale levels $p_{i}(1\leq i\leq \psi_{+})$ satisfying $\sum_{i=1}^{\psi_{+}}p_{i}=1$.

Let $S_{-}$ and $S_{+}$ represent the Shannon entropy of the left and right histograms, respectively. According to the definition of Shannon entropy, the formulas for calculating $S_{-}$ and $S_{+}$ are given as follows:(1)$$S_{-}=-\sum_{i=1}^{\psi_{-}}q_{i}\log_{2}q_{i},$$(2)$$S_{+}=-\sum_{i=1}^{\psi +}p_{i}\log_{2}p_{i}.$$

Let S represent the total Shannon entropy of the left and right parts, with the calculation formula as follows:(3)$$S=-\sum_{i=1}^{\psi_{-}+\psi +}{ο}_{i}\log_{2}{ο}_{i},$$
where:(4)$${ο}_{i}=\left\{\begin{array}{l} \frac{\mu_{-}}{\mu_{-}+\mu_{+}}q_{i}\;\;(1\leq i\leq \psi_{-})\\ \frac{\mu_{+}}{\mu_{-}+\mu_{+}}p_{i}\;(\psi_{-}+1\leq i\leq \psi_{-}+\psi_{+}). \end{array}\right.$$

Let $S_{\delta }=S_{+}-S$ represent the Shannon entropy difference between $S_{+}$ and S, where $S_{+}$ represents the Shannon entropy of the right histogram, and S represents the total Shannon entropy of the left and right parts. The following four propositions are used for analysis and argumentation: To maximize the Shannon entropy difference $S_{\delta }$, what distribution characteristics should the grayscale histogram in Figure 1 possess?

**Proposition** **1.**$S_{\delta }=\lambda \left(S_{+}-S_{-}\right)+\lambda \log_{2}\lambda +(1-\lambda )\log_{2}(1-\lambda )$*, where* $\lambda =\frac{\mu_{-}}{\mu_{-}+\mu_{+}}$.

**Proof.** 

(5)
$$\begin{array}{ll} S & =-\sum_{i=1}^{\psi_{-}+\psi +}{ο}_{i}\log_{2}{ο}_{i}\\ & =-\sum_{i=1}^{\psi_{-}}\frac{\mu_{-}}{\mu_{-}+\mu_{+}}q_{i}\log_{2}\frac{\mu_{-}}{\mu_{-}+\mu_{+}}q_{i}-\sum_{i=1}^{\psi_{+}}\frac{\mu_{+}}{\mu_{-}+\mu_{+}}p_{i}\log_{2}\frac{\mu_{+}}{\mu_{-}+\mu_{+}}p_{i}\\ & =-\lambda \sum_{i=1}^{\psi_{-}}q_{i}\log_{2}\lambda q_{i}-(1-\lambda )\sum_{i=1}^{\psi_{+}}p_{i}\log_{2}(1-\lambda )p_{i}\\ & =-\lambda (\sum_{i=1}^{\psi_{-}}q_{i}\log_{2}q_{i}+\sum_{i=1}^{\psi_{-}}q_{i}\log_{2}\lambda )-(1-\lambda )\left[\sum_{i=1}^{\psi_{+}}p_{i}\log_{2}p_{i}+\sum_{i=1}^{\psi_{+}}p_{i}\log_{2}(1-\lambda )\right]\\ & =-\lambda \sum_{i=1}^{\psi_{-}}q_{i}\log_{2}q_{i}-\lambda \log_{2}\lambda -(1-\lambda )\sum_{i=1}^{\psi_{+}}p_{i}\log_{2}p_{i}-(1-\lambda )\log_{2}(1-\lambda ) \end{array}.$$

Substituting Equations (1) and (2) into Equation (5), we obtain:(6)$$S=\lambda S_{-}-\lambda \log_{2}\lambda +(1-\lambda )S_{+}-(1-\lambda )\log_{2}(1-\lambda ).$$Therefore, $S_{\delta }=S_{+}-S$ can be represented as follows:(7)$$S_{\delta }=S_{+}-S=\lambda \left(S_{+}-S_{-}\right)+\lambda \log_{2}\lambda +(1-\lambda )\log_{2}(1-\lambda ).$$ □

**Proposition** **2.***When* $\lambda >\frac{1}{1+2^{S_{+}-S_{-}}}$*,* $S_{\delta }$ *is a monotonically increasing function with respect to*  λ.

**Proof.** The partial derivative of $S_{\delta }$ with respect  λ is given by:(8)$$S_{+}-S_{-}+\log_{2}\frac{\lambda }{1-\lambda }.$$Let the expression (8) be greater than 0, then the inequality solution for λ can be obtained: $\lambda >\frac{1}{1+2^{S_{+}-S_{-}}}$. According to the relationship between the derivative of a function and its monotonicity, it can be inferred that when $\lambda >\frac{1}{1+2^{S_{+}-S_{-}}}$, $S_{\delta }$ is a monotonically increasing function with respect to λ. □

**Proposition** **3.**$S_{\delta }$ *is a monotonically increasing function with respect to*  $S_{+}$.

**Proof.** The partial derivative of $S_{\delta }$ with respect to $S_{+}$ is λ and it is clear that λ>0. Consequently, $S_{\delta }$ is a monotonically increasing function of $S_{+}$. □

**Proposition** **4.**$S_{\delta }$ 
*is a monotonically decreasing function with respect to* $S_{-}$.

**Proof.** The partial derivative of $S_{+}-S$ with respect to $S_{-}$ is −λ, and it is clear that −λ<0. Thus, $S_{\delta }$ is a monotonically decreasing function of $S_{-}$. □

Propositions 1 to 4 indicate that, under the condition of $\lambda >\frac{1}{1+2^{S_{+}-S_{-}}}$, an increase in the values of λ and $S_{+}$, along with a decrease in the value of $S_{-}$, will lead to a greater value of $S_{\delta }$. Considering $\lambda =\frac{\mu_{-}}{\mu_{-}+\mu_{+}}$ and the relative magnitudes of $S_{+}$ and $S_{-}$, it can be further deduced that a relatively larger $\mu_{-}$ or a relatively smaller $\mu_{+}$, along with a relatively larger $S_{+}$ or a relatively smaller $S_{-}$ will result in a greater value of $S_{\delta }$. $\mu_{-}$ and $\mu_{+}$ represent the total frequencies of the left and right parts of the histogram, respectively. Thus, $S_{+}$ for the right part, with fewer pixels, should be as large as possible, whereas $S_{-}$ for the left part, with more pixels, should be as small as possible, which will result in a greater value of $S_{\delta }$. Based on the above analysis, a grayscale histogram with the following distribution characteristics will facilitate a greater value of $S_{\delta }$: the left histogram should contain a relatively larger number of pixels, and it should be as concentrated as possible within a narrower range of grayscales; conversely, the right histogram should contain a relatively smaller number of pixels, and it should be as dispersed as possible across a broader range of grayscales.

## 3. Maximizing Shannon Entropy Difference for Multiscale Multiplication Image

For a grayscale image f, the gradient magnitude image corresponding to the Gaussian filter scale σ can be calculated using the following equation:(9)$$A_{\sigma }(x,y)=||f*\nabla G||,$$
where $G(x,y;\sigma )=\frac{1}{\sqrt{2\pi }\sigma }e^{-\frac{x^{2}+y^{2}}{2\sigma^{2}}}$, the symbols ∇ and ∗ represent the operations of differentiation and convolution, respectively.

The multiscale multiplication transform of f is defined as the product of k different images $A_{\sigma_{i}}(x,y)$, resulting in a multiscale multiplication image A(x,y), expressed as:(10)$$A(x,y)=\prod_{i=1}^{k}A_{\sigma_{i}}(x,y).$$

According to the sampling theory of kernel function, the result of convolving a discrete Gaussian kernel of size (8σ+1)×(8σ+1) with an image can sufficiently approximate the result of convolving the complete Gaussian distribution with the same image. Furthermore, when performing convolution on digital images, the neighborhood size defined by the kernel is typically an odd number, such as 3×3, 5×5, 7×7, etc. Based on these two points, to generate a discrete convolution kernel of size (2i+1)×(2i+1), one can set $\sigma_{i}=0.25\times i(i\geq 1)$.

The computational complexity of the multiscale multiplication transform is determined by two sequential operations: gradient magnitude computation and pixel-wise multiplication. For an input image of size N×M and k value, the gradient computation at each scale $\sigma_{i}$ involves convolving the image with a Gaussian kernel of size $S_{i}\times S_{i}$, where $S_{i}=\lfloor 8\sigma_{i}+1\rfloor$. This operation has a complexity of $\Theta \left(NMS_{i}^{2}\right)$ per scale. The subsequent multiplication of k gradient images (Equation (10)) requires $\Theta \left(kNM\right)$ operations. Thus, the total complexity is $\Theta \left(\sum_{i=1}^{k}NMS_{i}^{2}+kNM\right)$.

Under the condition that the grayscales of the multiscale multiplication image A are normalized to [0,255], as the number of images k gradually increases, the response product of noise (or random details) will tend toward 0 (please refer to the second row of images in Figure 2b–f). In contrast, the product of the edge response will be distributed within [0,255]. Consequently, the grayscale histogram of A will exhibit the following characteristics: ① A right-heavy tail distribution; ② The mode gradually shifts to the left and equals 0 when k is sufficiently large; ③ As the value of k continues to increase beyond a certain critical value, the multiscale multiplication effect will cause more edge response values to approach 0, resulting in a sparser grayscale distribution within [0,255].

The relationship between the grayscale histogram of multiscale multiplication image A and the k value indicates that an appropriate k value needs to be found to make the grayscale histogram of image A better align with the desired distribution characteristics described in Section 2. To achieve this, the objective can be formulated as maximizing the Shannon entropy difference of the grayscale histogram of image A, which is formally expressed as:(11)$$(k^{*},l^{*})=\arg\underset{k\in Z^{+}}{\max}\underset{l\in \left[0,255\right]}{\max}\left(S_{+}-S\right).$$

Once the number of images $k^{*}$ participating in the multiscale multiplication transformation is determined, the corresponding multiscale multiplication image $A^{*}$ can be computed using Equations (9) and (10).

## 4. Constructing a New Shannon Entropy Objective Function

For an input grayscale image f, thresholding it with a grayscale value t produces a binary image $b_{t}$. From this binary image, the inner and outer contours can be extracted (Please refer to Step 4 in Section 6 for details). Pixels with a value of 1 in $c_{t}$ are used to sample the multiscale multiplication image $A^{*}$ calculated in Section 3. The grayscale histogram $H_{t}$ then can be reconstructed using the sampled pixels. Figure 3 provides a visual illustration of the aforementioned processing steps.

Assuming the grayscale histogram $H_{t}$ consists of $\psi_{t}$ grayscales, the discrete probability distribution of grayscales is denoted as $r_{i}(1\leq i\leq \psi_{t})$ with $\sum_{i=1}^{\psi_{t}}r_{i}=1$. According to the definition of Shannon entropy, the Shannon entropy corresponding to the grayscale histogram $H_{t}$ is as follows:(12)$$S\left(H_{t}\right)=-\sum_{i=1}^{\psi_{t}}r_{i}\log_{2}r_{i}.$$
Here, $S\left(H_{t}\right)$ is referred to as the Shannon entropy objective function.

For a grayscale image containing multiple objects, the grayscale differences among pixels within each object are generally relatively small. Consequently, in the multiscale multiplication image $A^{*}$, the pixels located within the objects are expected to exhibit relatively low grayscales. The grayscale histogram $H_{t}$, composed of these intra-object pixels, will have its mode skewed toward the left end, while the right side will appear sparse. This distribution characteristic of the grayscale histogram $H_{t}$ results in a lower Shannon entropy objective function value.

On the other hand, the edge pixels between different objects generally exhibit a relatively greater difference in grayscales. Consequently, in the multiscale multiplication image $A^{*}$, pixels located at the edges tend to have relatively high grayscales. The grayscale histogram $H_{t}$, comprised of these edge pixels, will have a wide grayscale distribution across the range [0,255], resulting in a relatively higher Shannon entropy objective function value.

According to the Canny edge detection theory, the gradient magnitude of ideal edge pixels represents local maxima [29]. In the constructed Shannon entropy objective function, the grayscale values corresponding to these maxima indicate the edge pixels that separate different objects. The thresholds derived from these maxima can effectively distinguish between objects and likely represent suitable segmentation thresholds (see the black dots on the Shannon entropy objective function in Figure 4). However, some maxima may result from image noise or random details, making them unsuitable as segmentation thresholds (see the red dots on the Shannon entropy objective function in Figure 4). The following section, Section 5, will combine local contour Shannon entropy to select multiple appropriate thresholds from all potential candidates.

## 5. Selecting Multiple Thresholds Based on Composite Local Contour Shannon Entropy

For a grayscale image, let the set P represent the grayscales corresponding to all the local maxima in the constructed Shannon entropy objective function:(13)$$P=\left\{t_{p}|\forall t\in \left\{t_{p}-1,t_{p},t_{p}+1\right\}\cap \left\{0,1,2,\ldots ,255\right\},\;S(t_{p})\geq S(t)\right\}.$$

Assuming the final number of thresholds to be selected is n, it is then necessary to choose n appropriate $t_{p}$ from the set P to serve as the thresholds. For clarity, these n thresholds are denoted as $t_{p_{1}},t_{p_{2}},t_{p_{3}},\ldots ,t_{p_{n}}$. With the grayscale image f and these n thresholds, n+1 inner and outer contour images can be constructed using Equation (14):(14)$$\begin{array}{l} c_{t_{1}}⇐b_{t_{1}}=\left\{\begin{array}{l} \;1\;\;\;\;f(i,j)<t_{p_{1}}\\ \;0\;\;\;\;\mathrm{otherwise}\; \end{array}\right.\\ c_{t_{2}}⇐b_{t_{2}}=\left\{\begin{array}{l} \;1\;\;\;\;t_{p_{1}}\leq f(i,j)<t_{p_{2}}\\ \;0\;\;\;\;\mathrm{otherwise}\; \end{array}\right.\\ \;\ldots \;\;\;\;\;\;\;\ldots \\ c_{t_{n}}⇐b_{t_{n}}=\left\{\begin{array}{l} \;1\;\;\;\;t_{p_{n-1}}\leq f(i,j)<t_{p_{n}}\\ \;0\;\;\;\;\;\;\mathrm{otherwise}\; \end{array}\right.\\ c_{t_{n+1}}⇐b_{t_{n+1}}=\left\{\begin{array}{l} \;1\;\;\;\;f(i,j)\geq t_{p_{n}}\\ \;0\;\;\;\;\mathrm{otherwise}\; \end{array}\right. \end{array},$$
where the symbol ⇐ represents the extraction of inner and outer contours from the binary image. The n+1 contour images will reflect different objects in the grayscale image, respectively. For example, the contour images $c_{t_{1}}\sim c_{t_{4}}$ in Figure 4 show the background, flower, rabbit, and man, respectively.

Given n thresholds $t_{1},t_{2},\ldots ,t_{n}$, and by thresholding the image f with each threshold to generate n binary images, these binary images form a set $\left\{b_{t}\right\}$. For each binary image $b_{t}$, the inner and outer contours are extracted (Please refer to Step 3 in Section 6 for details), resulting in a corresponding set of inner and outer contour images $\left\{c_{t}\right\}$.

**Proposition** **5.***For any* $t_{1},t_{2},\ldots ,t_{n}$*, the inner and outer contour images *$c_{t}$ 
*are always contained within the union of all the other inner and outer contour images, and when *
n=1
*, they are subsets of each other, that is:*
(15)$$c_{t_{n}}\subseteq \underset{n\neq k}{\cup }c_{t_{k}}.$$

**Proof.** Let $c_{in,t}$ represent the inner contour of  $b_{t}$, and $c_{out,t}$ represent the outer contour of $b_{t}$. Obviously, the inner and outer contour image $c_{t}$ satisfies the relationship $c_{t}=c_{in,t}\cup c_{out,t}$ with $c_{in,t}$ and $c_{out,t}$. The inner contour $c_{in,t}$ refers to the inner boundary pixels of the objects in the binary image $b_{t}$. These pixels have a value of 1 in $b_{t}$, and at least one of their neighboring pixels has a value of 0. The outer contour $c_{out,t}$ is obtained by extracting the inner contour from the complement image $\bar{b_{t}}$ of the binary image $b_{t}$. Thus, $c_{out,t}$ represents the outer boundary pixels of the objects in the binary image $b_{t}$.For any pixel $\alpha \in c_{in,t}$, if its position is represented by the coordinates $\left(x_{\alpha },y_{\alpha }\right)$, then $b_{t_{n}}\left(x_{\alpha },y_{\alpha }\right)=1$ holds and there exists at least one pixel coordinate $\left(x^{\prime },y^{\prime }\right)\in N\left(\alpha \right)$ such that $b_{t_{n}}\left(x^{\prime },y^{\prime }\right)=0$, where $N\left(\alpha \right)$ represents the four-neighborhood of α. In all $b_{t_{k}}(k\neq n)$, every pixel α satisfies $b_{t_{k}}\left(x_{\alpha },y_{\alpha }\right)=0$, and there exists a unique image $b_{t_{k}}^{*}$ such that the coordinates $\left(x^{\prime },y^{\prime }\right)$ satisfies $b_{t_{k}}^{*}\left(x^{\prime },y^{\prime }\right)=1$. Then, for the complement binary image $\bar{b_{t_{k}}^{*}}$, it follows that $\bar{b_{t_{k}}^{*}}\left(x_{\alpha },y_{\alpha }\right)=1$ and $\bar{b_{t_{k}}^{*}}\left(x^{\prime },y^{\prime }\right)=0$. Thus, $\alpha \in c_{out,k}$.Similarly, for any pixel $\alpha \in c_{out,t}$, if its position is represented by the coordinates $\left(x_{\alpha },y_{\alpha }\right)$, then $b_{t_{n}}\left(x_{\alpha },y_{\alpha }\right)=0$ holds, which implies $\bar{b_{t_{n}}}\left(x_{\alpha },y_{\alpha }\right)=1$, and there exists at least one pixel coordinate $\left(x^{\prime },y^{\prime }\right)\in N\left(\alpha \right)$ such that $\bar{b_{t_{n}}}\left(x^{\prime },y^{\prime }\right)=0$. In all $b_{t_{k}}(k\neq n)$, the coordinates $\left(x^{\prime },y^{\prime }\right)$ satisfy $b_{t_{k}}\left(x^{\prime },y^{\prime }\right)=0$, and there exists a unique image $b_{t_{k}}^{*}$ such that the coordinates $\left(x_{\alpha },y_{\alpha }\right)$ satisfies $b_{t_{k}}^{*}\left(x_{\alpha },y_{\alpha }\right)=1$. Thus, $\alpha \in c_{in,k}$.In summary, if a pixel $\alpha \in c_{in,t}$, then $\alpha \in c_{out,k}$; similarly, if a pixel $\alpha \in c_{out,t}$, then $\alpha \in c_{int,k}$. This means that $\alpha \in c_{t_{k}}$. Consequently, for any $t_{1},t_{2},\ldots ,t_{n}$, the inner and outer contour images $c_{t}$ are always contained within the union of all the other inner and outer contour images, and when n=1, they are subsets of each other. □

According to Proposition 5, if all inner and outer contour images are combined, at least one contour image will overlap entirely. Therefore, this study retains the contour image with the highest number of contour pixels as a standalone image (as shown in Figure 5 $c_{t_{\max}}$), excluding it from the combination process while merging the remaining contour images. This partial combination approach effectively reduces information overlap caused by merging all local contours, ensuring that the final composite contour image accurately reflects local features and actual complexity. Furthermore, it addresses issues of weight imbalance and local optimality that can arise when accumulating the Shannon entropy of all local contours, preventing certain local features from being overemphasized while others are neglected. The contour combination of the inner and outer contour images, excluding the image with the most contour pixels, is formally described(16)$$C_{t}=c_{t_{1}}\cup c_{t_{2}}\cup c_{t_{3}}\cup \ldots \cup c_{t_{n}}.$$

The pixels with values of 1 in $C_{t}$ and $c_{t_{\max}}$ are utilized to sample the multiscale multiplication image $A^{*}$ calculated in Section 3. The sampled pixels are then utilized to reconstruct the grayscale histograms $H_{t}$ and $H_{t_{\max}}$ (see Figure 5). Similarly, after performing the complement operation on $C_{t}$ and $c_{t_{\max}}$, two additional contour images, $\bar{C_{t}}$ and $\bar{c_{t_{\max}}}$ will be generated. The pixels with a value of 1 in $\bar{C_{t}}$ and $\bar{c_{t_{\max}}}$ are also utilized to sample the multiscale multiplication image $A^{*}$, and these sampled pixels are employed to reconstruct the grayscale histograms $H_{t}^{\prime }$ and $H_{t_{\max}}^{\prime }$.

Assuming the grayscale histograms $H_{t}$, $H_{t_{\max}}$, $H_{t}^{\prime }$, and $H_{t_{\max}}^{\prime }$ have $\psi_{t}$, $\psi_{t_{m}}$, $\psi_{t}^{\prime }$, and $\psi_{t_{m}}^{\prime }$ grayscales, respectively, with discrete probability distributions $r_{i}(1\leq i\leq \psi_{t})$, $s_{i}(1\leq i\leq \psi_{t_{m}})$, $u_{i}(1\leq i\leq \psi_{t}^{\prime })$, and $v_{i}(1\leq i\leq \psi_{t_{m}}^{\prime })$, where $\sum_{i=1}^{\psi_{t}}r_{i}=1$, $\sum_{i=1}^{\psi_{t_{m}}}s_{i}=1$, $\sum_{i=1}^{\psi_{t}^{\prime }}u_{i}=1$, and $\sum_{i=1}^{\psi_{t_{m}}^{\prime }}v_{i}=1$. According to the definition of Shannon entropy, the Shannon entropies corresponding to $H_{t_{\max}}$, $H_{t}^{\prime }$, and $H_{t_{\max}}^{\prime }$ are as follows:(17)$$S\left(H_{t_{\max}}\right)=-\sum_{i=1}^{\psi_{t_{m}}}s_{i}\log_{2}s_{i}.$$(18)$$S\left(H_{t}^{\prime }\right)=-\sum_{i=1}^{\psi_{t}^{\prime }}u_{i}\log_{2}u_{i}.$$(19)$$S\left(H_{t_{\max}}^{\prime }\right)=-\sum_{i=1}^{\psi_{t_{m}}^{\prime }}v_{i}\log_{2}v_{i}.$$

The objective function for selecting multiple thresholds based on the composite local contour Shannon entropy is as follows:(20)$$\left[t_{1}^{*},t_{2}^{*},\ldots ,t_{n}^{*}\right]=\underset{t_{1},t_{2},\ldots ,t_{n}\in P}{\arg\max}\left(\frac{S(H_{t})}{S(H_{t}^{\prime })}+\frac{S(H_{t_{\max}})}{S(H_{t_{\max}}^{\prime })}\right).$$

## 6. Algorithm Steps

Algorithm 1 describes the key steps for selecting the final multiple thresholds in the CLCSE method, while Figure 5 visually illustrates some key steps.
**Algorithm 1: CLCSE**Input: A grayscale image f and the number of thresholds n.Output: n thresholds $t_{1}^{*},t_{2}^{*},\ldots ,t_{n}^{*}$ and multi-level thresholding result image $f^{*}$Step 1: Compute the multiscale multiplication image $A^{*}$ using the method described in Section 3. For each grayscale $t\in \left[t_{\min},t_{\max}\right]$ in the input grayscale image f, repeat Steps 2 to 4 in ascending order of t. (Time complexity: dominated by the multiscale gradient computation, $\Theta \left(\sum_{i=1}^{k^{*}}NMS_{i}^{2}+k^{*}NM\right)$, where $k^{*}$ is the number of gradient images used in the multiplication, as analyzed in Section 3).Step 2: By thresholding the image f with t, the corresponding binary image $b_{t}$ is generated. (Time complexity: $\Theta \left(NM\right)$ per threshold, with L total thresholds, resulting in $\Theta \left(LNM\right)$).Step 3: Extract the inner and outer contour image $c_{t}$ from $b_{t}$, which can be broken down into three specific sub-steps. First, set all pixel values of $c_{t}$ to 1. Next, extract the inner contour: if a pixel in $b_{t}$ has a value of 1 and all its four-neighborhood pixels (defined as the top, bottom, left, and right adjacent pixels) are also 1, set the corresponding pixel in $c_{t}$ to 0. Finally, extract the outer contour: after performing a complement operation on $b_{t}$ to obtain $\tilde{b_{t}}$, the same judgement of pixels and their four-neighborhood is conducted on $\tilde{b_{t}}$, and $c_{t}$ is updated accordingly. (Time complexity: Contour extraction requires checking all pixels and their 4-neighbors, $\Theta \left(NM\right)$ per threshold).Step 4: Sample the multiscale multiplication image $A^{*}$ with pixels having a value of 1 in the image $c_{t}$, and construct a corresponding grayscale histogram $H_{t}$ using the sampled pixels. After normalizing the grayscale histogram to the range [0,255], calculate the corresponding Shannon entropy $S\left(H_{t}\right)$ from the normalized grayscale histogram. (Time complexity: Histogram construction and entropy calculation for each threshold require $\Theta \left(\left|C_{t}\right|\right)$, where $\left|C_{t}\right|\leq NM$, leading to an aggregate $\Theta \left(LNM\right)$).Step 5: After completing the loop calculations in Steps 2 to 4, utilize Equation (13) and the Shannon entropy objective function $S\left(H_{t}\right)$ to obtain the set P. Construct all subsets of P containing n thresholds, and identify the subset that maximizes the right-hand side of Equation (20), and set all n elements of this subset as the final thresholds $t_{1}^{*},t_{2}^{*},\ldots ,t_{n}^{*}$. (Time complexity: Threshold selection via sorting and peak detection over L candidates requires $\Theta \left(L\right)$).Step 6: Thresholding the image f using $t_{1}^{*},t_{2}^{*},\ldots ,t_{n}^{*}$, and output the thresholding result image $f^{*}$ and the thresholds $t_{1}^{*},t_{2}^{*},\ldots ,t_{n}^{*}$. (Time complexity: $\Theta \left(NM\right)$. This step involves iterating through each pixel and assigning it to a segmented region based on the selected thresholds, requiring constant-time operations per pixel).

## 7. Experimental Results and Discussions

### 7.1. Experimental Environment and Comparison Methods

The main software and hardware parameters used for testing are as follows: Intel Core i7-7700HQ CPU (2.80 GHz) (Intel Corporation, Santa Clara, CA, USA), 8 GB DDR4 memory (Samsung Electronics Co., Suwon, Republic of Korea), 64-bit Windows 10 operating system, and MATLAB R2012a as the integrated development environment. The test image set consists of six synthetic images, nine real-world images, and two widely recognized public datasets: PASCAL VOC 2007 and BSDS0500. The synthetic and real-world images, along with their corresponding ground truth, can be downloaded from https://wwtm.lanzouq.com/iVwkH2jf26oj (accessed on 15 May 2025). The PASCAL VOC 2007 dataset is available at http://host.robots.ox.ac.uk/pascal/VOC/voc2007/ (accessed on 15 May 2025), and BSDS0500 can be accessed via https://www2.eecs.berkeley.edu/Research/Projects/CS/vision/grouping/resources.html (accessed on 15 May 2025).

The proposed CLCSE method is compared with six recently developed multi-level thresholding methods: (1) multi-level thresholding using non-local means 2D histogram (2DNLM) [24]; (2) multi-level thresholding based on improved northern goshawk optimization (INGO) [22]; (3) an efficient Harris hawks-inspired multi-level thresholding method (HHO) [20]; (4) multi-level thresholding based on divergence measure and improved particle swarm optimization (IPSO) [19]; (5) exponential entropy multi-level thresholding using enhanced barnacle mating optimization (EBMO) [23]; and (6) multi-level thresholding using Q-learning-based golden jackal optimization (QLGJO) [21]. To ensure fairness in the comparison, the parameters for all the methods were set according to the recommendations provided by their respective authors.

### 7.2. Evaluation Metric

The Matthews Correlation Coefficient (MCC) [30,31] is used to evaluate the segmentation accuracy of the seven multi-level thresholding methods mentioned above. Compared to metrics such as Precision, Recall, Specificity, Dice, and Jaccard (also known as Intersection Over Union), the MCC is a more robust metric, as it yields a high value only when all classes are segmented accurately. The specific formulation of the MCC for multi-class scenarios is(21)$$MCC=\frac{c\times s-\sum_{k}^{K}p_{k}\times t_{k}}{\sqrt{\left(s^{2}-\sum_{k}^{K}p_{k}^{2}\right)\times \left(s^{2}-\sum_{k}^{K}t_{k}^{2}\right)}},$$
where $t_{k}=\sum_{i}^{K}C_{ik}$ represents the actual occurrence count of class k, $p_{k}=\sum_{i}^{K}C_{ki}$ represents the number of predictions for class k, $c=\sum_{k}^{K}C_{kk}$ indicates the total number of samples that have been correctly predicted, and $s=\sum_{i}^{K}\sum_{j}^{K}C_{ij}$ represents the total number of samples.

The MCC metric is used as the evaluation metric for multi-level thresholding, in contrast to traditional metrics such as the Structural Similarity Index (SSIM), Feature Similarity Index (FSIM), and Peak Signal-to-Noise Ratio (PSNR). The primary reason for this choice is that the MCC metric more accurately reflects the semantic information and segmentation accuracy in the thresholded images. The following experiment demonstrates that MCC is more robust than SSIM, FSIM, and PSNR.

Figure 6a shows a grayscale image degraded by Gaussian noise with a variance of 0.003, while Figure 6c,d present color visualizations of thresholding results under different thresholds. The SSIM, FSIM, and PSNR values for Figure 6c are 0.2798, 0.5697, and 19.1189, respectively, whereas for Figure 6d, the SSIM, FSIM, and PSNR values are 0.6827, 0.8275, and 20.0621, respectively. It is typically believed that higher SSIM, FSIM, and PSNR values indicate better multi-level thresholding results. However, the visual effect of Figure 6c is clearly superior to that of Figure 6d. Compared to these three traditional visual quality metrics, the MCC metric demonstrates greater robustness in reflecting semantic information and segmentation accuracy. The experimental results reveal that the images with better visual effects often exhibit higher MCC values. For instance, the MCC for Figure 6c is 0.9926, significantly surpassing the MCC of 0.2992 for Figure 6d.

### 7.3. Comparison Experiments on Synthetic Images

To evaluate the thresholding applicability of the CLCSE method and other methods, a test set consisting of six synthetic images with varying numbers of targets was generated. By constructing synthetic images with increasing complexity, ranging from single-target to multi-target scenarios, the study systematically evaluated the performance of the CLCSE method in multi-target, multi-level thresholding tasks. Figure 7 illustrates the Shannon entropy objective function (SEOF) curves used in the CLCSE method, along with the grayscale histograms of the synthetic images and the thresholds determined by the CLCSE method. Figure 8 and Table 1 present qualitative and quantitative comparison results of seven methods across six synthetic images.

Figure 7a is a grayscale image containing a single target, where the size ratio between the target and the background is severely imbalanced. In this case, the CLCSE method achieves perfect segmentation (MCC = 1.0000), while other methods fail to distinguish the target (MCC < 0.05). The success of CLCSE is rooted in its multiscale multiplication image selection mechanism (Section 3), which maximizes the Shannon entropy difference (Equation (11)) to suppress noise and enhance edge localization. Specifically, for Figure 7a, the optimal number of images $k^{*}=5$ participating in the multiscale multiplication transformation is determined by maximizing the entropy difference, and the final multiscale multiplication image is generated by multiplying gradient magnitudes from these five distinct scales (Equation (10)). This process effectively suppresses random noise (e.g., background texture in Figure 2a) while amplifying the target’s boundaries, enabling precise threshold selection even under extreme size imbalance. In contrast, metaheuristic methods (e.g., INGO, HHO) rely on global grayscale statistics (e.g., cross-entropy or inter-class variance) that assign equal weights to all pixels, causing thresholds to cluster around high-intensity regions (≈100) dominated by background pixels. Similarly, 2DNLM’s non-local spatial fusion introduces noise sensitivity, further degrading its performance (MCC = 0.0112). These results validate that maximizing Shannon entropy difference for multiscale multiplication images is critical for handling size-imbalanced segmentation tasks, whereas traditional methods lack such hierarchical edge discrimination.

For the grayscale image containing two targets shown in Figure 7b, all methods, except for the 2DNLM method, yielded satisfactory results. The 2DNLM method extends Rényi entropy to multi-level thresholding by combining a non-local means two-dimensional histogram with the exponential Kbest gravitational search method to determine the optimal thresholds. While this approach considers both pixel grayscale information and spatial information, its complex multi-information processing framework can lead to inferior performance compared to other methods in certain scenarios.

For the grayscale image with three targets shown in Figure 7c, the CLCSE method successfully achieved complete segmentation of all targets, while other methods exhibited inferior performance. The initial thresholds for these methods are commonly concentrated in the range of [0, 40], which can be attributed to the large number of pixels with a grayscale of 0. This phenomenon results in reduced frequency differences of other grayscales in this range, leading these methods to favor selecting lower grayscales during the initial threshold selection process. A similar phenomenon can be observed in Figure 7e, which reveals the strong dependence of these methods on the grayscale distribution. This reliance hampers their effectiveness in handling complex or non-uniform grayscale distributions, resulting in suboptimal segmentation performance.

In Figure 7d–f, as the number of targets in the grayscale image increases, the proposed CLCSE method maintains robust segmentation accuracy (MCC > 0.985), outperforming all compared methods. This success is rooted in the composite local contour Shannon entropy strategy (Section 5), which selectively combines local contours through a partial fusion approach to ensure precise edge localization for each target. Specifically, CLCSE retains the contour image with the highest number of contour pixels (e.g., the background in Figure 7d) as a standalone reference and combines the remaining contours using Equation (16). This partial combination strategy avoids redundancy caused by merging all local contours (Figure 5), ensuring that each retained contour accurately reflects the boundaries of distinct targets. For instance, in Figure 7f, retaining the largest contour (background) as a standalone reference while combining the remaining contours allows the method to focus on the subtle edges of individual targets, ensuring that the composite contour accurately reflects the true boundaries of all targets, thereby achieving an MCC of 0.9858. In contrast, metaheuristic methods like QLGJO and HHO rely on fixed fitness functions (e.g., inter-class variance) that bias thresholds toward dominant grayscale regions, leading to misclassification of smaller or adjacent targets (QLGJO MCC = 0.4915). The 2DNLM method, despite incorporating spatial context through non-local means 2D histograms, amplifies noise in dense configurations (MCC = 0.7102), while IPSO’s divergence-based objective fails to adapt to increasing target complexity (MCC = -0.0277). These results demonstrate that the partial contour fusion strategy in CLCSE, guided by composite local contour entropy, is essential for multi-target segmentation, as it adaptively preserves critical edges while suppressing redundant information, a capability absent in parameter-dependent frameworks.

### 7.4. Comparison Experiments on Real-World Images

To further demonstrate the potential applications of the proposed CLCSE method in various real-world scenarios, we tested and compared seven multi-level thresholding methods across nine representative images. These images were selected from diverse application domains, including material non-destructive testing, brain tumor MRI imaging, street-view thermal infrared imaging, satellite remote sensing for oil spill detection, ground scene remote sensing, infrared thermography for circuit breaker fault detection, steel surface defect inspection, and ship target monitoring.

Figure 9 presents the threshold selection results of the CLCSE method applied to nine real-world images. Figure 9a shows an industrial micro-defect image that requires precise differentiation of subtle differences due to the extremely small target and complex background texture. Figure 9b illustrates a brain tumor MRI image characterized by highly variable tumor morphologies and blurry edges, demanding robust recognition of shapes and boundaries. Figure 9c depicts an industrial product image with blurred shadows, where the method must maintain high performance despite noise and uncertain boundary conditions. Figure 9d presents a steel surface defect image that features low background contrast and unclear defect boundaries, requiring effective handling of low-contrast images. Figure 9e shows an infrared thermography image of circuit breaker faults, which necessitates the ability to detect minimal temperature differences due to a uniform grayscale distribution. Figure 9f highlights a ship monitoring image, where the grayscales of ocean waves are similar to those of the hull, requiring strong discrimination capabilities to avoid mis-segmentation. Figure 9g provides an oil spill detection remote sensing image that poses challenges of uneven target-to-background ratios and complex ocean textures, necessitating effective segmentation of imbalanced scales. Figure 9h presents a ground remote sensing image, where intricate grass textures and random details increase the difficulty of target area extraction. Figure 9i illustrates a street-view thermal infrared image, where people appear small and their grayscale is similar to that of the background, necessitating that the method effectively handles small targets with precision.

Figure 10 and Figure 11 present the thresholding results obtained by the seven methods shown in Figure 9, while Table 2 summarizes the corresponding MCC values of these methods on nine real-world images. The results indicate that the CLCSE method achieves relatively higher MCC values on most test images, demonstrating superior thresholding adaptability and robustness in addressing challenges such as complex backgrounds, low contrast, blurry boundaries, and imbalanced size ratios. In comparison, the 2DNLM method consistently exhibits lower MCC values on all test images, reflecting its weaker thresholding adaptation in these complex scenarios. The QLGJO method performs well on test images in Figure 9b,d,e, achieving MCC values exceeding 0.8500, which indicates that the QLGJO method has specific advantages in scenarios characterized by low contrast. Additionally, the EBMO method has higher MCC values on images Figure 9a,f,g, than other methods, except for CLCSE, indicating its relatively strong adaptability and robustness in handling tasks involving complex backgrounds and imbalanced target-to-background ratios. For INGO, HHO, and IPSO methods, their MCC values exhibit obvious fluctuations across these test images, indicating that their thresholding accuracy is not consistently stable across different scenarios.

### 7.5. Extensive Evaluation on Benchmark Datasets

To comprehensively validate the robustness and generalization capability of the proposed CLCSE method, we conducted additional experiments on two widely recognized public datasets: PASCAL VOC 2007 and BSDS0500. The experimental results can be broadly categorized into two types. The first type comprises images with well-defined targets and minimal grayscale overlap between distinct objects (e.g., the six representative examples in Figure 12). For this category, the CLCSE method achieved superior performance compared to the six other thresholding methods (Figure 13), demonstrating the highest MCC values as quantified in Table 3. The second type involves images lacking clear targets and exhibiting substantial grayscale overlap between objects (e.g., the three challenging cases in Figure 14). For such scenarios, all seven thresholding methods performed poorly (Figure 15), with consistently low MCC values as summarized in Table 4. These findings highlight the method’s effectiveness in scenarios with distinguishable targets while underscoring the inherent limitations of grayscale-based thresholding approaches in complex, low-contrast environments.

### 7.6. Comparison of Computation Efficiency

Even with identical hardware and software parameters, different methods may exhibit slight fluctuations in CPU runtime when applied to the same image. To minimize these variations, each method was executed 20 times on the same image, and the average runtime was calculated as the mean CPU runtime. As shown in Table 5, the computational efficiency of the proposed CLCSE method is validated against six state-of-the-art approaches. CLCSE achieves the highest efficiency on synthetic images (mean runtime: 0.5663 s) and demonstrates competitive performance on real-world images (mean runtime: 1.1324 s), showcasing its superiority over metaheuristic methods. This efficiency advantage primarily arises from the direct selection of candidate thresholds using local maxima in the Shannon entropy objective function (Section 4), which eliminates the need for iterative optimization.

Unlike metaheuristic methods (e.g., INGO, QLGJO), which require population initialization and hundreds of iterations to explore the search space, CLCSE directly extracts candidate thresholds from the local maxima of the Shannon entropy curve (Equation (13)). This process eliminates the need for stochastic parameter tuning and reduces computational complexity from $\Theta \left(NT\right)$ (for metaheuristics) to $\Theta \left(L\right)$, where L represents the number of grayscale levels. For example, in Figure 4, the entropy curve displays distinct peaks that correspond to semantic boundaries (e.g., target–background transitions), allowing CLCSE to identify optimal thresholds in a single pass. In contrast, methods like IPSO and EBMO iteratively evaluate all possible threshold combinations (e.g., 150 iterations in QLGJO), resulting in substantial runtime penalties (IPSO: 4.8638s vs. CLCSE: 1.1324 s on real-world images).

### 7.7. Discussions

The proposed CLCSE method fundamentally diverges from existing multi-level thresholding techniques through its edge-driven entropy optimization framework, which directly selects thresholds based on local maxima of composite contour Shannon entropy, bypassing iterative metaheuristic optimization. Unlike conventional methods (e.g., 2DNLM, HHO, QLGJO) that rely on a two-stage paradigm—designing objective functions (e.g., cross-entropy, divergence measures) and deploying population-based algorithms to search for thresholds—CLCSE integrates a multiscale multiplication transform to enhance critical edges while suppressing noise, and dynamically constructs contour-guided entropy models to identify thresholds from semantically meaningful boundaries. This eliminates parameter dependency, e.g., population size and mutation rates. While baseline methods suffer from premature convergence in texture-rich or low-contrast scenarios due to global grayscale statistics, CLCSE prioritizes localized edge distributions, achieving superior accuracy in target-distinct cases.

Many existing approaches, including 2DNLM, INGO, HHO, IPSO, EBMO, and QLGJO, predominantly follow a “black-box optimization” framework, where thresholds are derived by maximizing/minimizing predefined objective functions (e.g., Otsu’s variance, exponential entropy) via metaheuristic algorithms. While these methods demonstrate theoretical potential, their practical performance is constrained by intrinsic limitations in objective function design and extrinsic bottlenecks in metaheuristic optimization.

(1)Intrinsic Limitations

The predefined objective functions used by baseline methods often fail to adapt to complex scenarios due to their reliance on global statistical measures or rigid mathematical formulations:

The 2D Rényi entropy objective function in the 2DNLM method combines non-local spatial information with grayscale distributions. While it improves noise robustness compared to 1D methods, its computational complexity ($O\left(N^{2}M^{2}\right)$) becomes prohibitive for high-resolution images. Additionally, the fixed Rényi parameter (α=0.45) limits adaptability to images with varying texture complexity. The symmetric cross-entropy objective function in the INGO method assumes balanced foreground and background distributions. In size-imbalanced cases (e.g., small defects in industrial images), this function biases thresholds toward dominant regions (e.g., backgrounds), leading to misclassification. The Kullback–Leibler divergence-based cross-entropy in the HHO method measures global similarity between the original and segmented images. However, it struggles with low-contrast boundaries (e.g., blurred edges in medical images), where grayscale overlap between regions creates ambiguous fitness landscapes. The arithmetic–geometric divergence criterion in the IPSO method focuses on minimizing statistical discrepancies between regions. While effective for homogeneous textures, it fails to capture edge information, resulting in fragmented segmentation for images with complex structures. The exponential entropy in the EBMO method’s objective function replaces logarithmic terms with exponential gains to avoid undefined values. However, this modification amplifies noise sensitivity, as seen in low-SNR images. Otsu’s inter-class variance in the QLGJO method maximizes separability between regions but ignores spatial coherence. For images with overlapping intensity distributions, this leads to oversegmentation.

(2)Extrinsic Limitations

The 2DNLM method exhibits three primary shortcomings. Firstly, the computation of its 2D histogram requires analyzing relationships between pixel values and non-local means, significantly increasing complexity compared to 1D histograms. This process involves calculating and statistically aggregating neighborhood pixels for every image pixel, resulting in substantial computational overhead. Secondly, the method is highly sensitive to parameter settings. Its enhanced gravitational search algorithm (eKGSA) relies on empirically tuned parameters (e.g., population size, gravitational constant G, iteration limits, see Table 6), which lack theoretical guidance and risk suboptimal convergence or local minima traps. Additionally, the Rényi entropy parameter α, fixed at 0.45 in experiments, limits adaptability across diverse images despite its stability within α ∈ [0.1, 0.9]. Thirdly, the method struggles with noise robustness. While non-local means filtering mitigates noise to some extent, it fails to eliminate strong or structurally complex noise, often blurring critical details and distorting the 2D histogram. Consequently, residual noise alters threshold distributions, leading to misclassification or blurred boundaries in segmentation results.

The INGO method suffers from two critical drawbacks. Firstly, the integration of multiple enhancement strategies—including cubic chaotic initialization, best-worst reverse learning, and lens imaging reverse learning—significantly increases algorithmic complexity. This structural intricacy raises implementation challenges and necessitates the adjustment of additional parameters, where improper configurations may degrade performance. Secondly, the method faces hyperparameter selection difficulties (see Table 6). Key parameters, such as the cubic chaos control parameter (ρ=2.595) and lens imaging scaling factor (m), lack systematic guidance for optimal tuning. In practice, users must empirically determine these values through extensive experimentation, which complicates usability and limits adaptability to diverse datasets.

The HHO method exhibits two key limitations. Firstly, its performance heavily depends on image feature characteristics. The method’s reliance on grayscale distributions makes it sensitive to uneven intensity variations or noise interference, leading to inaccurate boundary detection in homogeneous regions. Additionally, its minimum cross-entropy criterion, based solely on grayscale information, fails to capture structural or chromatic details in complex images (e.g., color-rich or texture-dense scenes), resulting in suboptimal segmentation. Secondly, the algorithm is prone to local optima stagnation, a common issue in population-based optimization algorithms. In complex segmentation tasks with large search spaces, the Harris hawks’ convergence strategy prioritizes exploitation over exploration, causing premature convergence to suboptimal thresholds and degrading accuracy.

The IPSO method faces four primary limitations. Firstly, its robustness to noise remains underexplored, despite the prevalence of noise in practical applications such as crack detection. Sensitivity to noise can distort grayscale distributions, leading to inaccurate threshold selection. Secondly, the algorithm requires manual tuning of multiple parameters, including learning factors ($c_{1},c_{2}$), population size, and maximum iterations (see Table 6). The absence of systematic guidelines for parameter optimization restricts its adaptability across diverse datasets. Thirdly, while local stochastic perturbations are introduced to mitigate local optima, the algorithm still risks stagnation in high-dimensional or complex search spaces, particularly when handling intricate image data. Lastly, the method primarily relies on grayscale intensity distributions for thresholding, neglecting critical spatial features such as texture and edge information. This oversimplification limits segmentation accuracy in scenarios where structural or contextual cues are essential.

The EBMO method exhibits two critical limitations in algorithmic performance. Despite incorporating a Gaussian mutation strategy and random flow steps to enhance exploration, the algorithm remains susceptible to local optima when addressing high-dimensional or multi-modal optimization problems, particularly with complex fitness landscapes. This limitation arises from insufficient global search capability, preventing it from identifying true global thresholds in intricate scenarios. Additionally, the method struggles to balance convergence speed and precision. While improvements in convergence rate are achieved, this often occurs at the expense of segmentation accuracy, especially in tasks demanding extremely high precision, such as fine-grained medical image analysis.

The QLGJO method exhibits three primary limitations. Firstly, it demonstrates significant parameter sensitivity, requiring meticulous tuning of key hyperparameters such as the Q-learning rate (λ), discount factor (γ), and mutation strategy coefficients (see Table 6). Suboptimal parameter settings can slow convergence or trap the algorithm in local optima, necessitating extensive experimental calibration in practical applications. Secondly, despite integrating reinforcement learning and mutation mechanisms to enhance population diversity, the algorithm remains prone to local optima stagnation in complex, multi-modal optimization landscapes. This issue stems from insufficient exploration capability in later iterations, where diminished diversity restricts global search effectiveness. Lastly, the method’s robustness is inconsistent, particularly under data variations or noise interference. While it achieves stable performance in controlled experiments, real-world scenarios with subtle intensity shifts or artifacts may degrade segmentation reliability. Collectively, these limitations highlight challenges in balancing adaptability, precision, and computational stability.

## 8. Conclusions and Future Work

In recent years, research on multi-thresholding methods has primarily focused on rapid solutions using metaheuristic optimization algorithms. The proposed CLCSE method introduces a novel objective function that facilitates the direct search for multiple optimal thresholds, providing a fresh perspective on image multi-level thresholding. The CLCSE method employs multiscale multiplication images along with composite inner and outer contour images to jointly construct a one-dimensional grayscale histogram. Shannon entropy serves as the model for entropy calculation, and local contour Shannon entropy is subsequently integrated for the automatic selection of multiple thresholds. Synthetic images and real-world images from various application fields are utilized as test images, with the MCC metric employed for performance evaluation. Experimental results demonstrate that the proposed CLCSE method outperforms the 2DNLM, INGO, HHO, IPSO, EBMO, and QLGJO multi-level thresholding methods in terms of MCC. Additionally, the comparison of average CPU runtimes for each method indicates that the proposed CLCSE method is efficient and comparable to the most computationally efficient method. Overall, the CLCSE method offers significant advantages in segmentation accuracy and robustness without requiring complex parameter tuning.

The proposed CLCSE method faces limitations in efficiency when processing large images. Specifically, the logical operations performed on binary images during the contour combination process are the primary factors influencing CPU runtime, with computational costs increasing as image size grows. Future research will focus on reducing the computational cost of these logical operations on binary images to enable quicker attainment of multiple optimal thresholds. Additionally, the experimental results in Section 7.5 highlight the challenges of CLCSE, as well as six other multi-level thresholding methods, in handling images with ambiguous targets and significant grayscale overlaps. To address this limitation, future work could consider applying regional consistency processing to the images, such as integrating superpixel-based preprocessing. With superpixel-based preprocessing, the local homogeneity and spatial coherence within superpixels may enhance the robustness of threshold selection in low-contrast or texture-rich scenarios.

## Figures and Tables

**Figure 1 entropy-27-00544-f001:**
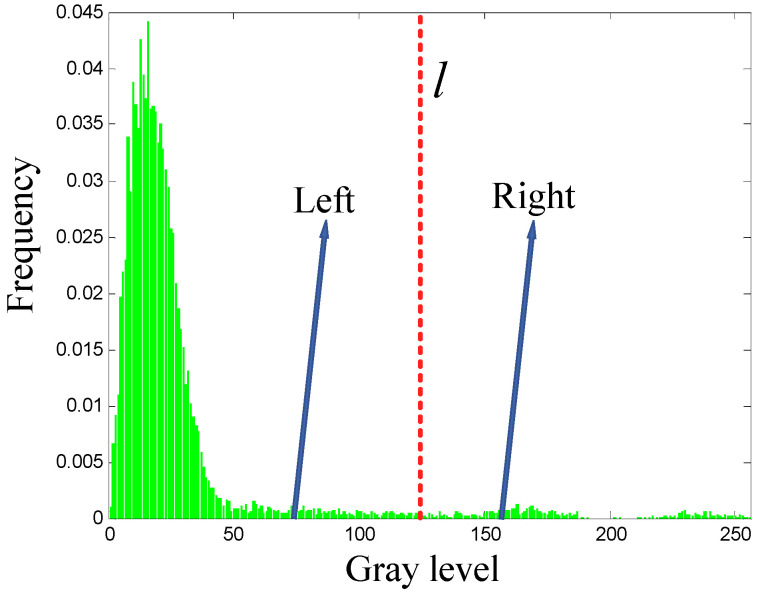
Schematic diagram illustrating the division of a grayscale histogram into left and right sections.

**Figure 2 entropy-27-00544-f002:**
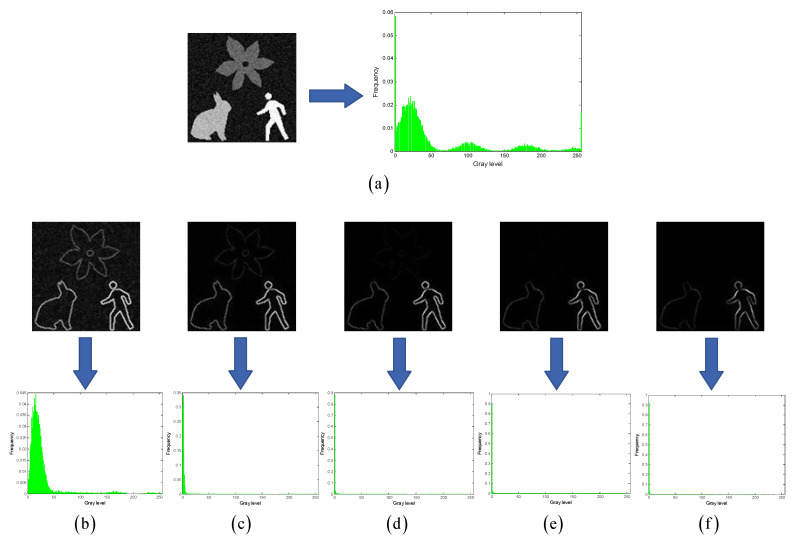
Multiscale multiplication effect. (**a**) shows an original grayscale image and its grayscale histogram; (**b**–**f**) show multiscale multiplication images and their corresponding grayscale histograms for k values of 1, 2, 3, 4, and 5, respectively.

**Figure 3 entropy-27-00544-f003:**
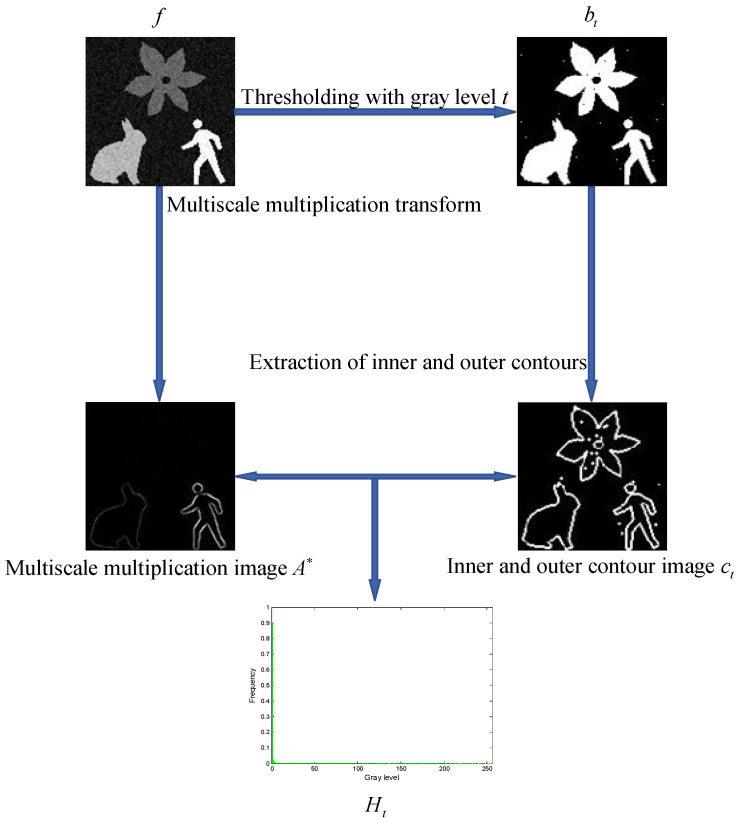
An intuitive illustration of the key steps involved in constructing the Shannon entropy objective function.

**Figure 4 entropy-27-00544-f004:**
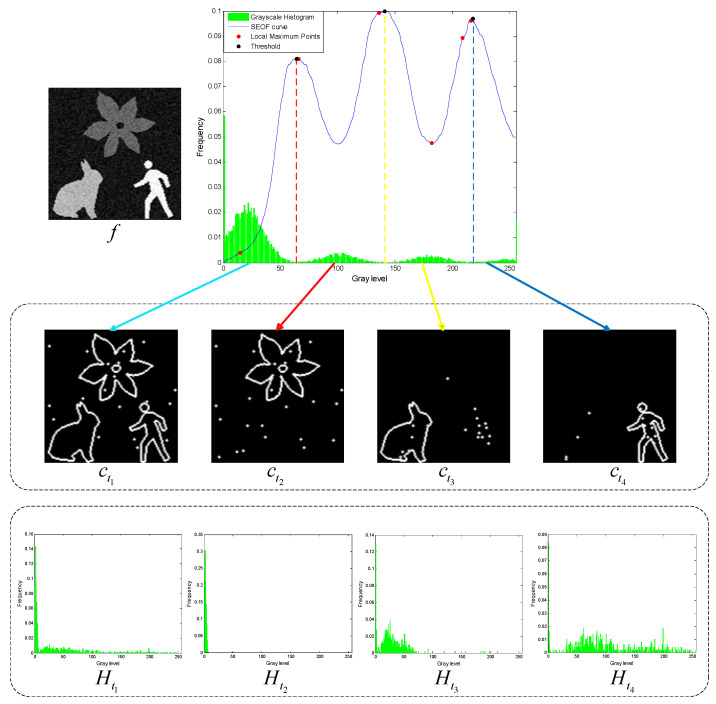
The Shannon entropy objective function (SEOF) curve, along with inner and outer contour images obtained at different thresholds, and the grayscale histograms reconstructed from the multiscale multiplication image $A^{*}$.

**Figure 5 entropy-27-00544-f005:**
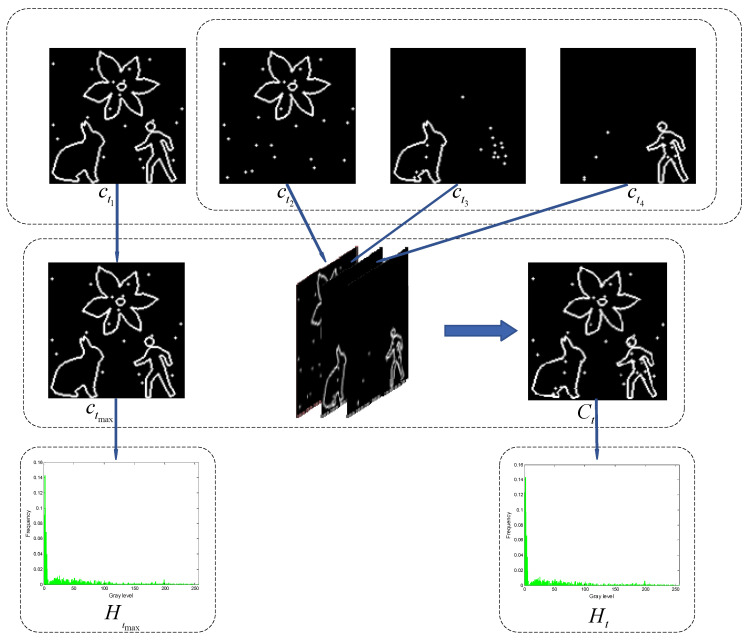
Key steps for selecting multiple thresholds based on composite local contour Shannon entropy.

**Figure 6 entropy-27-00544-f006:**
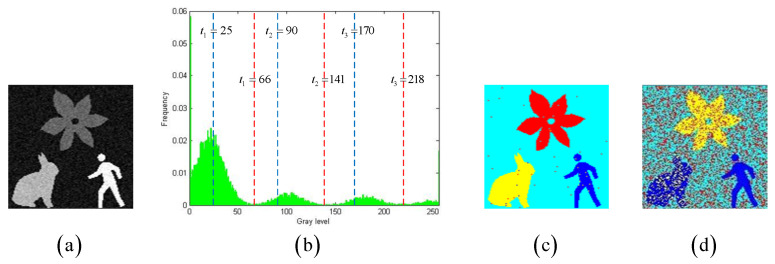
(**a**) shows an original grayscale image, (**b**) shows its grayscale histogram with two groups of thresholds. (**c**) presents the color visualization results for one group of thresholds at 66, 141, and 218, (**d**) displays the color visualization results for another group of thresholds at 25, 90, and 170.

**Figure 7 entropy-27-00544-f007:**
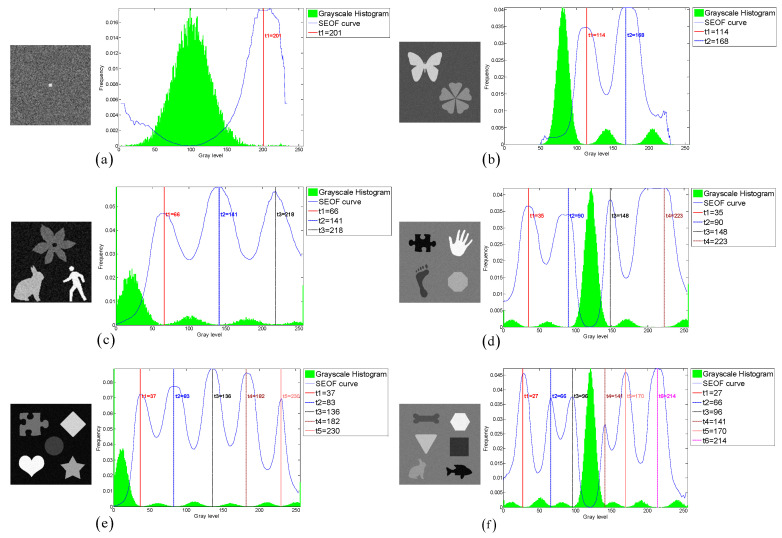
(**a**–**f**) Threshold selection using the CLCSE method for synthetic images with varying numbers of targets.

**Figure 8 entropy-27-00544-f008:**
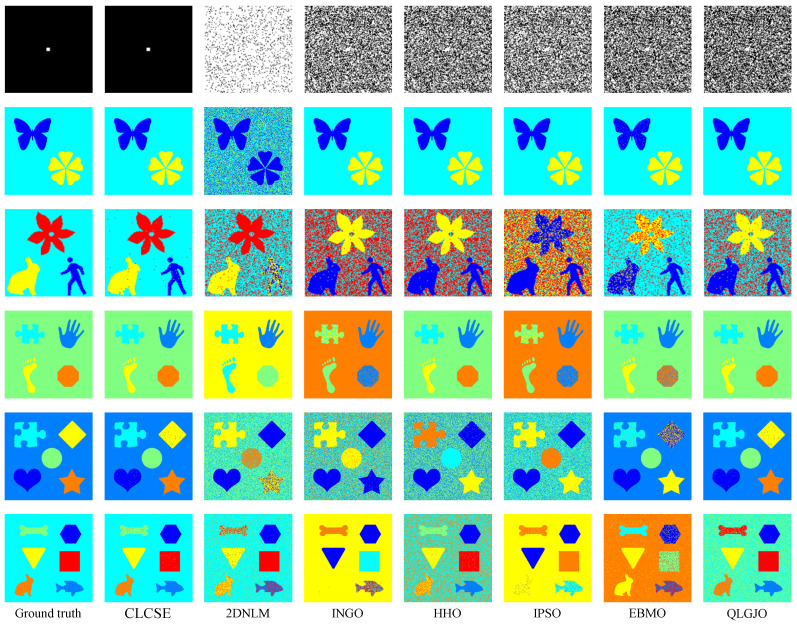
Thresholding results of different multi-leveling methods on synthetic images.

**Figure 9 entropy-27-00544-f009:**
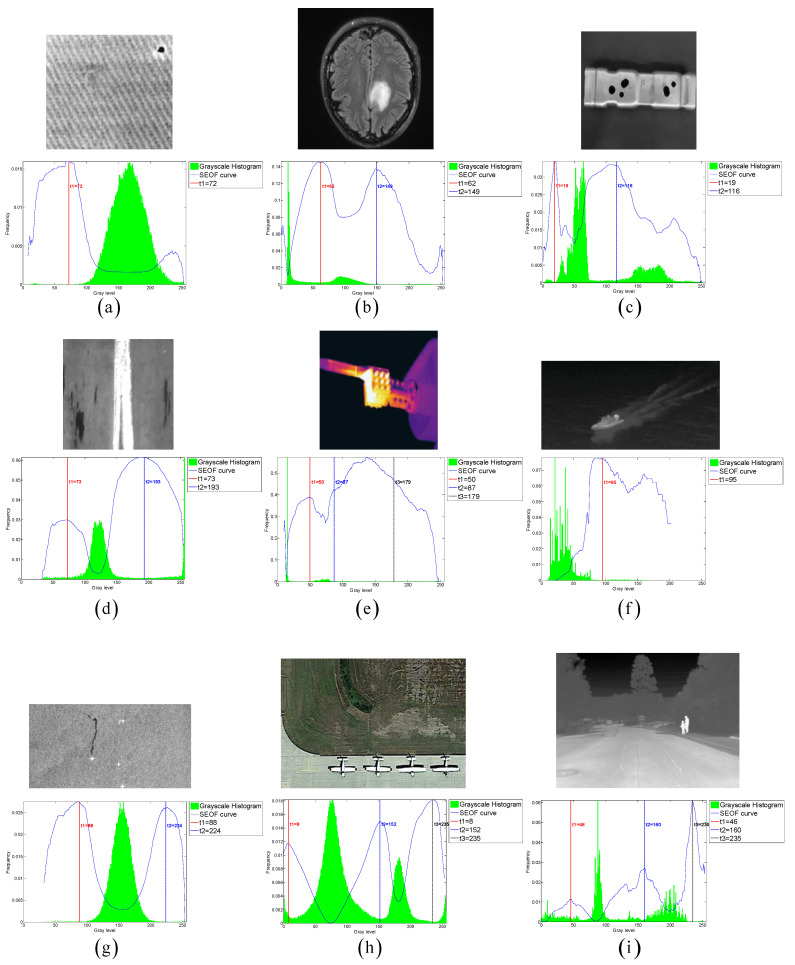
(**a**–**i**) Threshold selection using the CLCSE method on nine real-world images.

**Figure 10 entropy-27-00544-f010:**
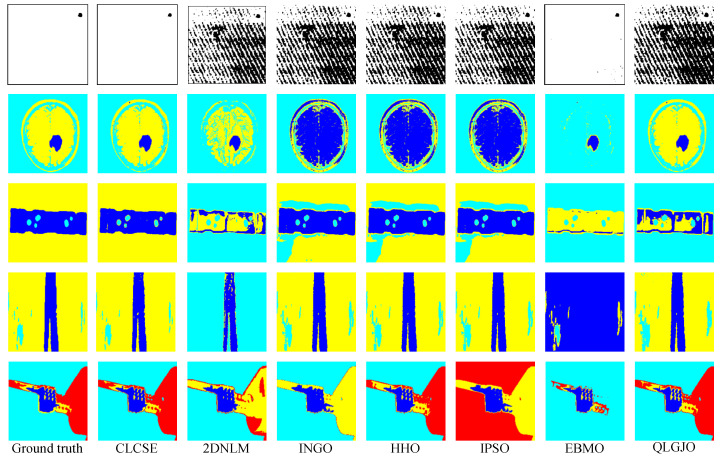
Thresholding results of different methods on Figure 9a–e.

**Figure 11 entropy-27-00544-f011:**
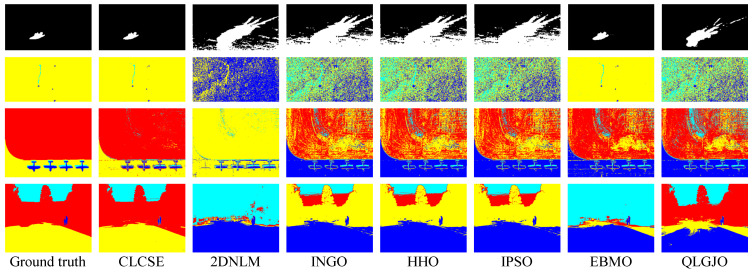
Thresholding results of different methods on Figure 9f–i.

**Figure 12 entropy-27-00544-f012:**
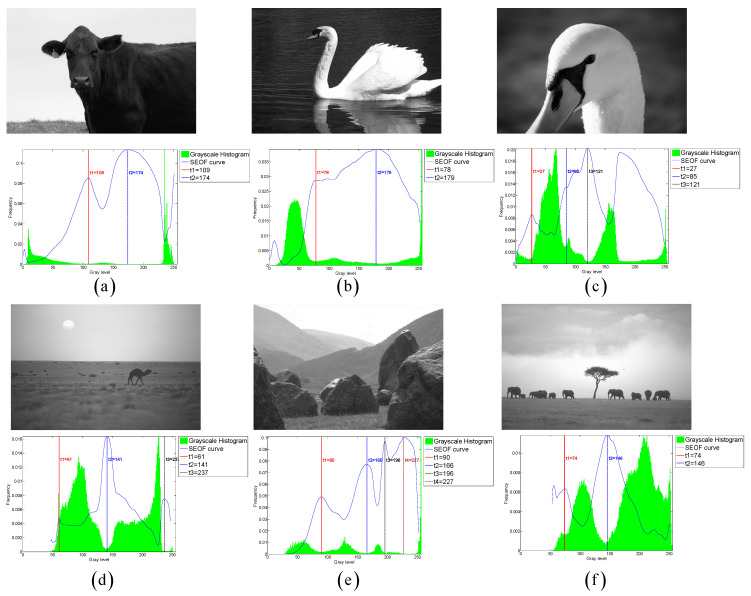
(**a**–**f**) Threshold selection using the CLCSE method on six images from the PASCAL VOC 2007 and BSDS0500 datasets.

**Figure 13 entropy-27-00544-f013:**
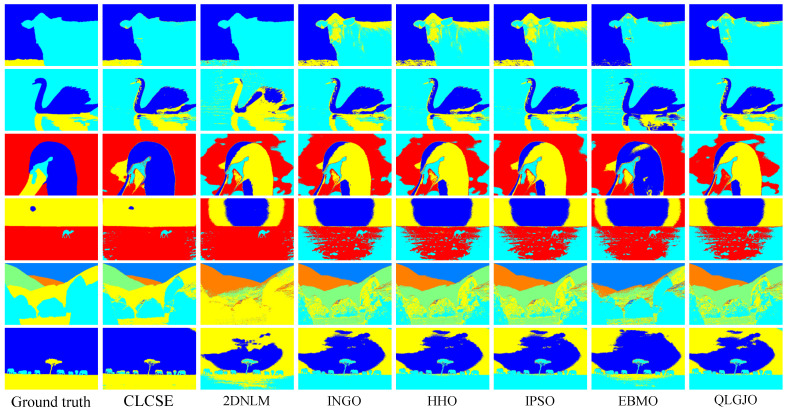
Thresholding results of different methods on Figure 12a–f.

**Figure 14 entropy-27-00544-f014:**
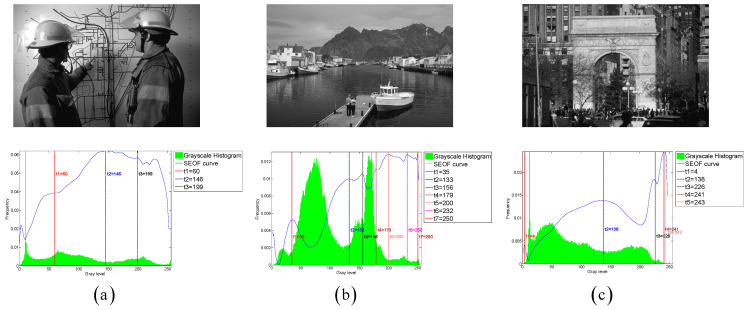
(**a**–**c**) Threshold selection using the CLCSE method on three images from the PASCAL VOC 2007 and BSDS0500 datasets.

**Figure 15 entropy-27-00544-f015:**
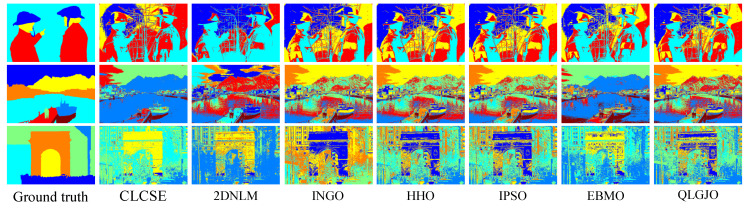
Thresholding results of different methods on Figure 14a–c.

**Table 1 entropy-27-00544-t001:** MCC values of seven multi-level thresholding methods on six images in Figure 8. Bold values denote the highest MCC.

Test Images	CLCSE	2DNLM	INGO	HHO	IPSO	EBMO	QLGJO
Figure 7a	**1.0000**	0.0112	0.0347	0.0357	0.0335	0.0382	0.0407
Figure 7b	**0.9997**	0.2879	**0.9997**	0.9995	**0.9997**	0.9947	**0.9997**
Figure 7c	**0.9926**	0.6335	0.0710	0.0975	0.0137	0.6942	0.1882
Figure 7d	**0.9983**	0.0643	−0.0493	0.9956	−0.3353	0.9485	0.9971
Figure 7e	0.9964	0.1027	0.0499	0.1900	0.1014	0.8242	**0.9966**
Figure 7f	**0.9858**	0.7102	−0.0869	0.5977	−0.0277	0.1056	0.4915

**Table 2 entropy-27-00544-t002:** MCC values for seven multi-level thresholding methods on nine real-world images. Bold values denote the highest MCC.

Test Images	CLCSE	2DNLM	INGO	HHO	IPSO	EBMO	QLGJO
Figure 9a	**0.8419**	0.0491	0.0491	0.0491	0.0523	0.7765	0.0452
Figure 9b	**0.9820**	0.7486	0.4019	0.3958	0.3927	0.2406	0.9742
Figure 9c	**0.9841**	−0.1513	0.7512	0.7493	0.7601	0.3949	0.0419
Figure 9d	**0.9867**	−0.1314	0.9078	0.9078	0.9078	0.0156	0.8785
Figure 9e	**0.9836**	0.6562	0.5690	0.9497	−0.1777	0.4537	0.9660
Figure 9f	**0.9725**	0.1582	0.1402	0.1402	0.1355	0.8906	0.4054
Figure 9g	**0.9549**	0.3770	0.0745	0.0769	0.0745	0.8554	0.0740
Figure 9h	**0.9126**	0.1154	0.1431	0.1544	0.1296	0.2736	0.2309
Figure 9i	**0.9921**	0.1222	−0.1637	−0.1659	-0.1637	0.3811	0.5274

**Table 3 entropy-27-00544-t003:** MCC values of seven multi-level thresholding methods on six images in Figure 12. Bold values denote the highest MCC.

Test Images	CLCSE	2DNLM	INGO	HHO	IPSO	EBMO	QLGJO
Figure 12a	**0.9792**	0.9224	0.8089	0.7933	0.8089	0.8741	0.9296
Figure 12b	**0.** **8910**	0.6961	0.8712	0.8760	0.8753	0.8020	0.8618
Figure 12c	**0.** **8330**	0.3167	0.3850	0.3202	0.4713	0.7894	0.2271
Figure 12d	**0.9** **568**	0.5373	0.3990	0.3936	0.3990	0.3507	0.3853
Figure 12e	**0.** **8908**	0.2253	0.2802	0.2768	0.2802	0.5248	0.3563
Figure 12f	**0.9** **446**	0.1267	0.1053	0.1065	−0.0050	0.2330	0.0711

**Table 4 entropy-27-00544-t004:** MCC values of seven multi-level thresholding methods on three images in Figure 14.

Test Images	CLCSE	2DNLM	INGO	HHO	IPSO	EBMO	QLGJO
Figure 14a	−0.1507	−0.1247	−0.0371	−0.0371	−0.0371	−0.1200	−0.0541
Figure 14b	0.0091	0.0306	−0.0258	−0.0230	−0.0242	−0.0400	−0.0351
Figure 14c	−0.0114	0.1085	−0.0989	−0.0237	−0.0237	0.0233	0.0154

**Table 5 entropy-27-00544-t005:** Comparison of CPU runtime for seven multi-level thresholding methods.

Method	CPU Runtime on Synthetic Images (s)		CPU Runtime on Real-World Images (s)	
	Mean	Standard Deviation	Mean	Standard Deviation
CLCSE	0.5663	0.1903	1.1324	0.8896
2DNLM	4.2720	2.4165	12.8299	3.6199
INGO	3.4071	0.4998	4.2230	1.5968
HHO	0.6295	0.0548	0.5952	0.0189
IPSO	3.5057	0.4735	4.8638	1.5119
EBMO	0.9244	0.3002	1.0648	0.3712
QLGJO	0.7861	0.2128	0.6465	0.1352

**Table 6 entropy-27-00544-t006:** Key features of seven multi-level thresholding methods.

Method Name	Objective Function	Swarm Intelligence Optimization Algorithm	Parameters (Excluding Threshold Count)	Hyperparameter Settings
CLCSE	Composite Local Contour Shannon Entropy	Not Applicable	0	None
2DNLM	2D Rényi Entropy	Enhanced Kbest Gravitational Search (eKGSA)	5 (Population Size, Max Iterations, Gconstant, Beta, Final_per)	α = 0.45 (fixed); Others: Empirical Values
INGO	Minimized Symmetric Cross-Entropy	Improved Northern Goshawk Optimization (INGO)	4 (Cubic Chaos ρ = 2.595, Lens Scaling Factor, Population Size, Max Iterations)	Scaling Factor (experimental); Others: Fixed
HHO	Minimized Cross-Entropy (KL Divergence)	Harris Hawks Optimization (HHO)	3 (Population Size, Max Iterations, Energy Decay Factor)	Energy Decay Factor (empirical)
IPSO	Minimum Arithmetic-Geometric Divergence	Improved Particle Swarm Optimization (IPSO)	6 (Population Size, c1/c2, Scaling Factor, Perturbation Probability, Max Iterations)	Manual Tuning Required
EBMO	Maximized Exponential Entropy	Enhanced Barnacle Mating Optimization (EBMO)	5 (Population Size, Max Iterations, Gaussian Mutation Mean/Std, Scaling Factor)	Gaussian Parameters (empirical)
QLGJO	Maximized Otsu’s Inter-Class Variance	Q-Learning Guided Golden Jackal Optimization (QLGJO)	7 (λ, γ, Mutation Parameters, Population Size, Max Iterations, Hybrid Mode Ratio, Lévy Flight)	λ and γ Require Fine-Tuning

## Data Availability

The test image set and its corresponding ground truth image set in this study are accessible at https://wwtm.lanzouq.com/iVwkH2jf26oj (accessed on 15 May 2025).

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
