# Peer review of "Multi-Level Thresholding Based on Composite Local Contour Shannon Entropy Under Multiscale Multiplication Transform"

_entropy, 2025, doi:10.3390/e27050544_

Round 1

Reviewer 1 Report (New Reviewer)

Comments and Suggestions for Authors

This paper introduces a novel multi-level thresholding technique (CLCSE) for image segmentation, which integrates Shannon entropy differences, multiscale multiplication transforms, and composite local contour information. Distinct from existing methods that predominantly depend on metaheuristic optimization algorithms, the proposed approach selects optimal multiscale multiplication images by maximizing the Shannon entropy difference and formulates a new entropy-based objective function through the dynamic combination of contour images. The method is rigorously evaluated against six recently proposed multi-level thresholding algorithms on both synthetic and real-world datasets. The experimental results demonstrate that CLCSE achieves superior segmentation accuracy, quantified by the Matthews Correlation Coefficient, and improved computational efficiency.

Nevertheless, I have several suggestions for the authors to strengthen the manuscript further:

1. Parameter Sensitivity Analysis: Although the authors assert that their method does not require complex parameter tuning, there is limited discussion regarding the sensitivity of the algorithm to specific parameters, such as the number of scales in the multiscale multiplication transform. Providing a detailed sensitivity analysis would help clarify the robustness and generalizability of the proposed approach.

2. Comparative Analysis Depth: While the proposed method is compared against several state-of-the-art techniques, the manuscript would benefit from a more in-depth analysis explaining why certain methods underperform in specific scenarios. Such a discussion would provide valuable insights into the strengths and limitations of the proposed and baseline methods, and help guide future research in this area.

Author Response

Reviewer 2 Report (New Reviewer)

Comments and Suggestions for Authors

Review of the article "Multi-level Thresholding Based on Composite Local Contour Shannon Entropy under Multiscale Multiplication Transform" The article proposes a novel multi-level thresholding approach based on composite local contour Shannon entropy (CLCSE) and a multiscale multiplication transform. The approach is a perfect fit to the MDPI Entropy journal's area of information-theoretic methods and entropy-based modeling. The authors contribute by tackling problems such as removing local optima and parameter setting. Nevertheless, the most important drawback of the research paper is the limited number of synthetic and real images used for validation.  The presented experimental section is not a good support of the introduced method. In the revised paper, a more extensive evaluation of benchmark datasets would be necessary to establish the generalizability and reliability of the suggested approach. The authors should fully rewrite the Experiments chapter.

The Introduction explains the latest methods for image segmentation using thresholding techniques. It describes the most important such articles in the state-of-the-art and their applications in various fields

The second section provides the mathematical background of Shannon entropy difference, which is used to determine the best thresholding levels. The theory is well-structured, with clear and well-understandable formulas and proofs.

The notation and formulation are clear, the mathematical background of the proposed method is easy understandable. But, the authors should discuss and reference other studies on similar entropy-based methods

The 3rd section explains the multiscale multiplication transform. The authors should analyze the computational complexity of this approach. A Θ (theta) notation analysis is required to compare its efficiency to methods such as Otsu’s method or Kapur’s entropy-based thresholding.

The paper introduces an entropy objective function using inner and outer contour sampling. Combining Shannon entropy with local contour analysis is a good idea. The authors have to present quantitative experiments (e.g., statistical validation) to show that the proposed objective function works also well for the segmentation of large datasets.

The 5th chapter summarizes the choice of several thresholds determined by composite local contour entropy. Dealing with noisy images that can generate stuck-in local maxima of the entropy function should have been studied more rigorously. Moreover, in real images, the contours are not easy to detect; there are a lot of false contour parts that have to be eliminated.

  1. Algorithm Implementation Algorithmic implementation of the CLCSE method is given step by step. A formal time complexity analysis (Θ notation) should be provided to determine its scalability.Hyperparameter selection (e.g., number of scales for multiscale multiplication) should be further discussed.
  2. Experimental Results and Discussions The algorithm is tested on six synthetic images and nine real images and compared against six state-of-the-art multi-level thresholding methods. Comparison based on the Matthews Correlation Coefficient (MCC) is adequate. The paper's most important limitation is its small dataset size (just 6 synthetic and 9 real-world images). This is the major reason the paper cannot be accepted in its current form. A thresholding algorithm must be tested reliably on benchmark datasets with hundreds of images. The authors must conduct some statistical tests to verify the significance of the improvements over the competing methods. The proposed method is a very good fit for the MDPI Entropy Journal, and the idea and mathematical foundations of the method are well explained. However, some serious weaknesses of the paper led to a major revision decision. These are the limited dataset size (6 synthetic and 9 real images) for the validation of the method, no testing on benchmark datasets, no computational complexity analysis (Θ notation), and statistical validation is missing and reduce the credibility and general applicability of the paper. Please correct all of these. So, my final decision is Accept with Major Revisions.

Round 2

Reviewer 2 Report (New Reviewer)

Comments and Suggestions for Authors

All my concerns from the first review were corrected.

This manuscript is a resubmission of an earlier submission. The following is a list of the peer review reports and author responses from that submission.

Round 1

Reviewer 1 Report

Comments and Suggestions for Authors

This paper proposes a Shannon entropy-based multi-level thresholding method using composite contours. Although the results show some advantages, there are still many problems. Specific comments are as follows.

1. This work and the comparison method of the experiment are too traditional, and the author should try to compare it with some of the latest methods.

2.The experimental scenario is not innovative, only some results are given but without any useful reasons.

3.The English presentation of the paper needs further improvement. 

Comments on the Quality of English Language

This paper proposes a Shannon entropy-based multi-level thresholding method using composite contours. Although the results show some advantages, there are still many problems. Specific comments are as follows.

1. This work and the comparison method of the experiment are too traditional, and the author should try to compare it with some of the latest methods.

2.The experimental scenario is not innovative, only some results are given but without any useful reasons.

3.The English presentation of the paper needs further improvement.